# Macroscopic homochiral helicoids self-assembled via screw dislocations

Shengfu Wu[1,2], Xin Song[1], Cong Du[1] & Minghua Liu [1,2] ✉

Chirality is a fundamental property in nature and is widely observed at hierarchical scales from subatomic, molecular, supramolecular to macroscopic and even galaxy. However, the transmission of chirality across different length scales and the expression of homochiral nano/microstructures remain challenging. Herein, we report the formation of macroscopic homochiral helicoids with ten micrometers from enantiomeric pyromellitic diimide-based molecular triangle (PMDI-Δ) and achiral pyrene via a screw dislocation-driven co-self-assembly. Chiral transfer and expression from molecular and supramolecular levels, to the macroscopic helicoids, is continuous and follows the molecular chirality of PMDI-Δ. Furthermore, the screw dislocation and chirality transfer lead to a unidirectional curvature of the helicoids, which exhibit excellent circularly polarized luminescence with large $|g_{lum}|$ values up to 0.05. Our results demonstrate the formation of a homochiral macroscopic organic helicoid and function emergence from small molecules via screw dislocations, which deepens our understanding of chiral transfer and expression across different length scales.

Chirality is a ubiquitous fundamental property in nature and plays a crucial role in chemistry, biology, materials science and physics[1–4]. In biological systems, chiral molecules connect sequentially to form macromolecules by selecting homochiral molecules and further fabricate higher-ordered helical nanostructures such as double-helical DNA and α-helical proteins[5,6], in which the chirality was transferred from molecules to higher-level structures with the conservation of molecular chirality[7–16]. In materials systems, it is found that two pathways can direct the formation of macroscopic chiral morphologies or shapes. One is the small chiral molecules such as amino acids, peptides and chiral ligands, which can be associated with selective growth of certain facets and the breaking of intrinsic crystal symmetry[17–27]. The other is the screw dislocations or chiral topological defects, which have been proven to be an important way of fabricating one-dimensional (1D)[28–34] and two-dimensional (2D)[35–40] materials with macroscopic chiral shapes. In particular, they will possibly cause the curvature in crystals and drive the formation of mesoscale twisted crystals[32] or supertwisted spiral crystals[41] on non-Euclidean surfaces[42]. In addition, screw dislocation is also a classical method for producing three-dimensional (3D) spirals from inorganic crystals and polymers[43]. However, 3D helicoid structures

rarely originate from small organic molecules. In macroscopic particle systems, the helical hexagonal crystals assembled by 1 μm long filamentous viruses via screw dislocations with the chiral transfer from particles to helical superstructures were found[44]. However, such cross-length scale transfer of macroscopic chirality[7] induced by screw dislocations in supramolecular systems is still relatively rare[37]. Especially, although many chiral structures such as helix and twist[7,8], have been reported, the formation of the nano/microstructure beyond Euclidean geometry[42] is still a great challenge.

Here, we propose a hierarchical self-assembly strategy based on screw dislocations to construct a macroscopic 3D helicoid with strong circularly polarized luminescence (CPL)[45–47]. The helicoid is the only ruled minimal surface other than the plane and has less been reported through self-assembly. The electron-deficient chiral pyromellitic diimide-based molecular triangle (PMDI-Δ) with rigid π-conjugated aromatic linkers was co-assembled with an achiral pyrene to form intermolecular charge transfer (CT) complexes[48–50] by a thermal annealing process (Figs. 1a-c). Helical twists emerged with pyrene sandwiching into adjacent PMDI-Δ via π-π stacking and CT interactions. The CT complexes grew into a curved microsheet as a subunit

[1]Beijing National Laboratory of Molecular Sciences (BNLMS) and CAS Key Laboratory of Colloid, Interface and Thermodynamics, Institute of Chemistry, Chinese Academy of Sciences, North First Street 2, Zhongguancun, Beijing 100190, China. [2]University of Chinese Academy of Sciences, No.19(A) Yuquan Road, Beijing 100049, China. ✉e-mail: liumh@iccas.ac.cn

due to helical growth around a screw dislocation. Layer-by-layer screw dislocations drove the formation of macroscopic homochiral helicoids via efficient chirality transfer (Fig. 1b). These helicoids showed a pronounced macroscopic helicity controlled by the molecular chirality of PMDI-Δ and exhibited excellent CPL properties. This work demonstrated the transmission and expression of chirality at molecular, supramolecular, and macroscopic levels.

## Results

### Self-assembly of homochiral helicoids

The rigid chiral PMDI-Δ macrocycle[51,52] (Fig. 1a and Supplementary Fig. 1) with electron deficiency was selected as an electron acceptor and the size-matched electron-rich pyrene (Pyr) was selected as an electron donor. In a typical co-assembly protocol (Supplementary Fig. 2), the PMDI-Δ and Pyr were dissolved in DMF by ultrasound to give

a transparent solution. Water was added as the anti-solvent to provide a homogeneous yellow suspension, which was then annealed at 140 °C for 10 minutes until the charge-transfer (CT) complex between the two molecules was broken and the yellow color of the suspension completely faded. The subsequent natural cooling of the suspension to room temperature facilitated the co-assembly, which could be distinctly observed with the naked eye due to dramatic color changes (Fig. 1c and Supplementary Fig. 3a). The self-assemblies of both PMDI-Δ and Pyr were white, whereas the co-assemblies presented a yellow color under natural light due to CT complex formation after annealing (Supplementary Fig. 3). The morphology of co-assemblies obtained after annealing was observed by scanning electron microscopy (SEM), in which unique micrometer-sized homochiral helicoids were visualized (Figs. 1d, e, h and Supplementary Fig. 4), while individual PMDI-Δ or Pyr could only form achiral nanostructures (Supplementary Fig. 5).

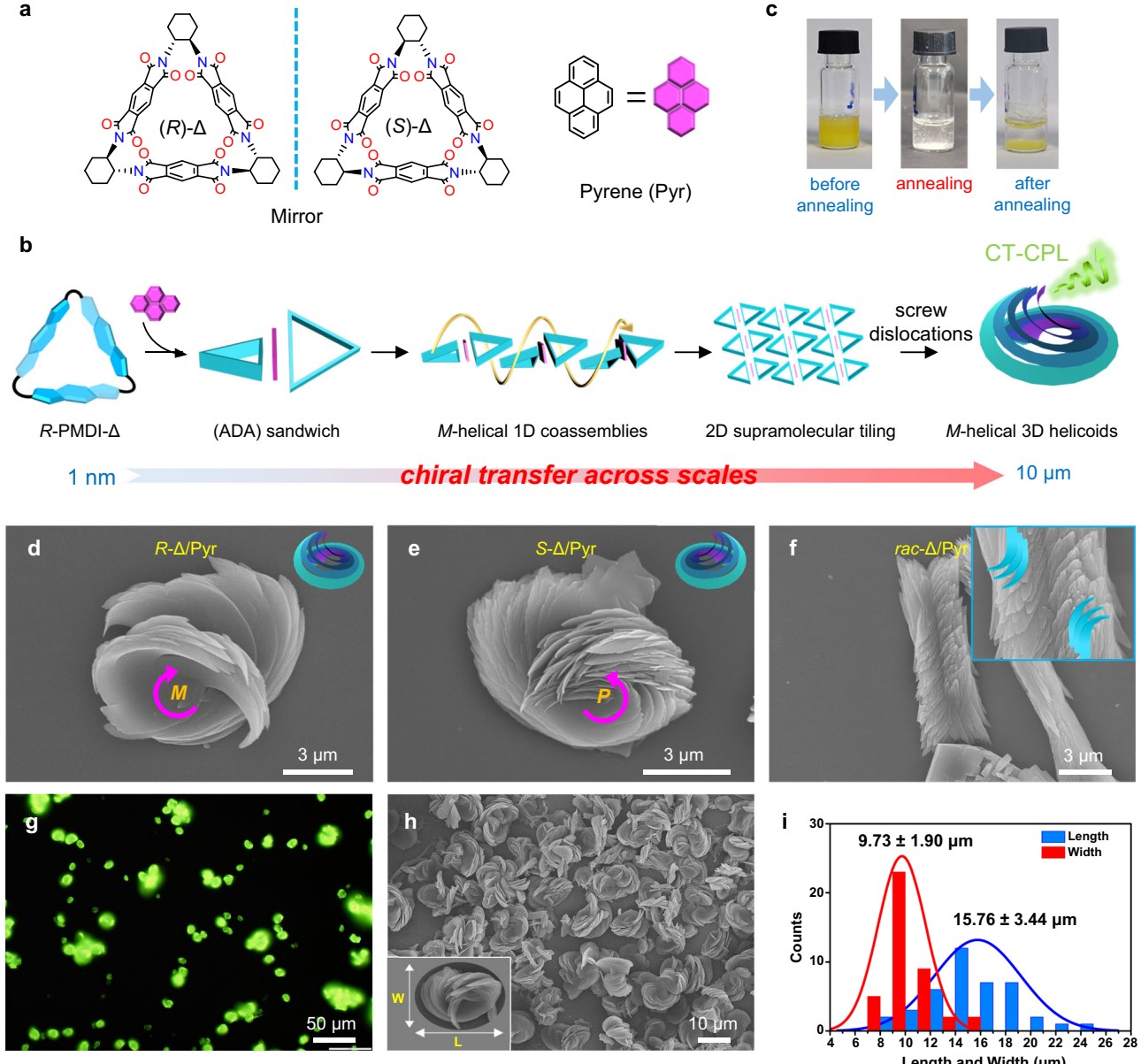

**Fig. 1 | Hierarchical co-self-assembly of homochiral helicoids. a** Molecular structures of *R*-PMDI-Δ, *S*-PMDI-Δ and Pyrene, respectively. **b** Transmission and expression of chirality across different length scales at molecular, supramolecular and macroscopic levels. **c** Photographs of sample vials in different assembly stages under natural light correspond to before annealing, annealing at 140 °C, and after annealing from left to right, respectively. **d** to **f** and **h** SEM images of (**d**) *R*-PMDI-Δ/Pyr helicoids[DMF], (**e** and **h**) *S*-PMDI-Δ/Pyr helicoids[DMF] and (**f**) *rac*-PMDI-Δ/Pyr co-assemblies[DMF]. **g** Fluorescence microscopy image of *R*-PMDI-Δ/Pyr helicoids[DMF]. **i** Statistical analysis of the length and width of helicoids[DMF].

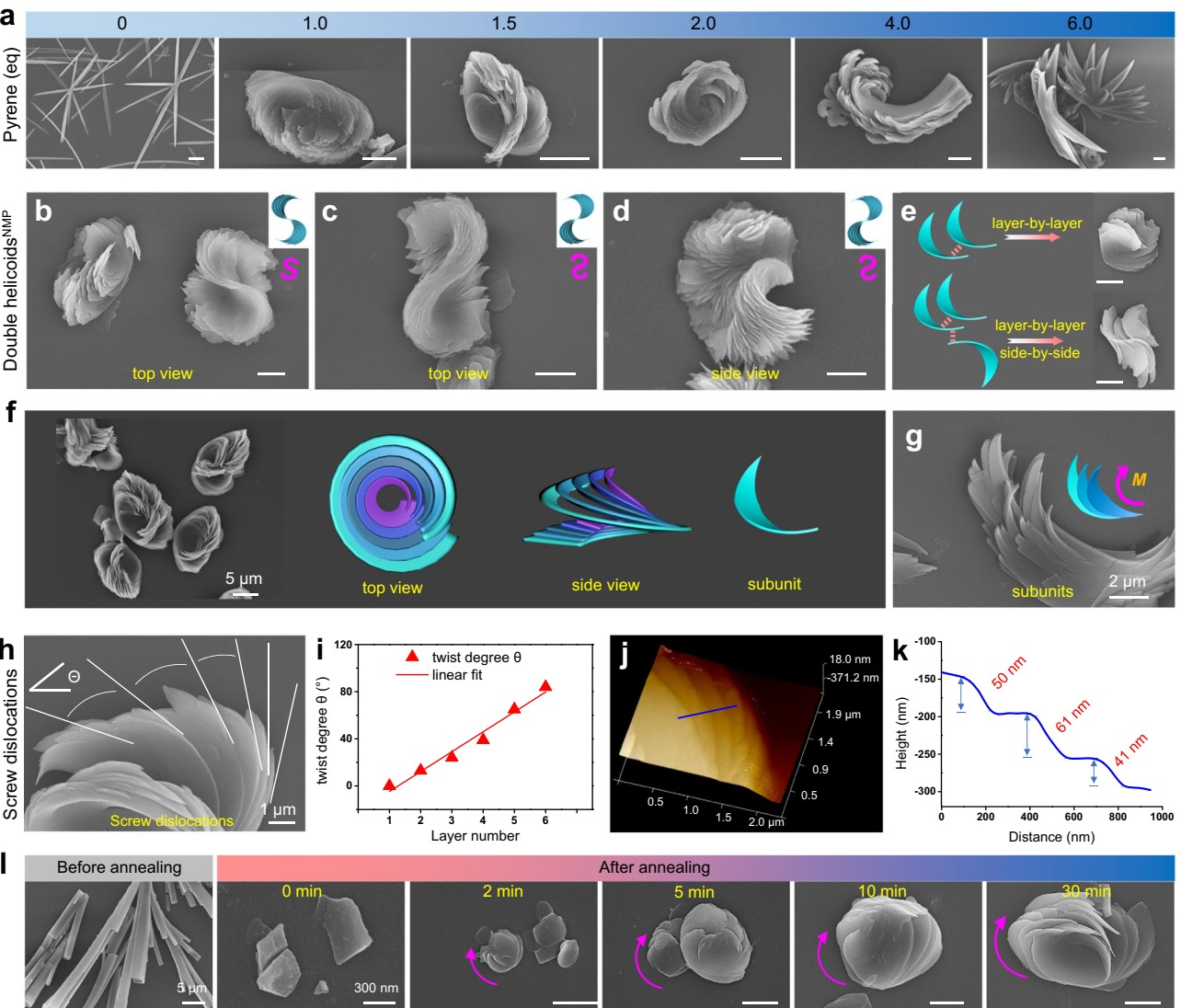

**Fig. 2 | Morphology evolution and structure analysis of homochiral helicoids.**
**a** Typical SEM images of *R*-PMDI-Δ/Pyr co-assemblies[DMF] with different molar equivalent Pyr. **b** SEM images of helicoids[NMP] and S-type double helicoids[NMP] (for *R*-Δ/Pyr). **c, d** SEM images of anti-S-shaped double helicoids[NMP] (for *S*-Δ/Pyr), from top view (**c**) and side view (**d**). **e** Formation mechanism of helicoids[NMP] and double helicoids[NMP]. **f, g** SEM images and corresponding cartoon representations of typical *R*-PMDI-Δ/Pyr helicoids[DMF] and its subunits. **h–k** Screw dislocations model analysis. **h** Magnified view of the edge region of the *R*-Δ/Pyr helicoids[DMF] shown in Fig. 1d, in which the tangents to the edges of layered subunits were highlighted with solid white line. **i** Scatter plot and linear fitting curve of the evolving twist angles $\theta$ in (**h**) as a function of layer number counted from the bottom to the top. The average of $\theta$ is equal to 16.9° and the coefficient of determination $R^2$ was equal to 0.97. **j** The 3D AFM morphology image of the edge of *R*-Δ/Pyr helicoid[DMF], as denoted by the red box in Supplementary Fig. 17a. **k** The height profile across the blue line in (**j**). **l** SEM images of *R*-PMDI-Δ/Pyr co-assemblies[DMF] before annealing and with different natural cooling times after annealing. Scale bars, 5 μm (**a**) and 2 μm (**b** to **e** and **l**).

The SEM images showed that the size of helicoids was relatively uniform (Fig. 1h). Statistical analysis gives an average length of 15.76 ± 3.44 μm and an average width of 9.73 ± 1.90 μm (Fig. 1i, Supplementary Fig. 6 and Supplementary Table 1). The fluorescence microscope images also revealed that the size of helicoids was about a dozen micrometers (Fig. 1g). Intriguingly, when *R*-PMDI-Δ and Pyr were employed, a left-handed (*M*-helical) helicoid was obtained (Fig. 1d and Supplementary Fig. 4a), while *S*-PMDI-Δ/Pyr co-assemblies showed a right-handed (*P*-helical) one (Fig. 1e and Supplementary Fig. 4b). This demonstrated that the macroscopic helicity of the helicoids was determined by the molecular chirality of PMDI-Δ. In comparison, when racemic PMDI-Δ and Pyr were co-assembled, it was observed that microsheets twisted in two opposite directions and were piled up on each other, indicating the self-sorting of the PMDI-Δ racemate triggered by Pyr (Fig. 1f and Supplementary Fig. 7). Morphological observations revealed that other electron donors with various sizes

different from Pyr could not co-assemble into any chiral nanostructures with PMDI-Δ, which suggested the necessity of size matching between PMDI-Δ and Pyr for the formation of helicoids (Supplementary Figs. 8 to 9). The amount of Pyr added and the ratio of DMF and $H_2O$ were found to affect the macroscopic morphology of the co-assemblies as well (Fig. 2a and Supplementary Fig. 10). It should be noted that while mixed solvents such as THF/$H_2O$ and dioxane/$H_2O$ could not afford the helicoids (Supplementary Fig. 11), the morphology of the co-assemblies obtained in NMP/$H_2O$ was similar to that obtained in DMF/$H_2O$ (Supplementary Fig. 12), except that a fully-formed S-type double helicoid (for *R*-PMDI-Δ/Pyr) and an anti-S-shaped double helicoid (for *S*-PMDI-Δ/Pyr) could also be observed in NMP/$H_2O$ (Figs. 2b to d and Supplementary Fig. 13). Interestingly, in DMF/$H_2O$, only two helicoids side by side were observed, but they could not be completely connected and grow to form a whole structure (Supplementary Fig. 14). By successfully capturing the initial morphology of helicoids

and double helicoids in NMP/H$_2$O, a possible formation mechanism for both types of helicoids was proposed (Fig. 2e and Supplementary Fig. 15). The process involved the layer-by-layer stacking of chiral subunits, which led to the formation of helicoids. However, some chiral microsheets could be connected side-by-side while stacking layer-by-layer, which led to the co-assemblies growing in two directions, thus forming a double helicoid structure by two helicoids connecting side-by-side. The formation of complete double helicoids in NMP/H$_2$O could be attributed to the lower polarity and the higher viscosity of NMP compared with DMF (Supplementary Table 2). These two factors enhanced intermolecular CT interactions between PMDI-Δ and Pyr and gave rise to a more stable dispersion of chiral subunits in the solution, which enabled the connection of side-by-side and facilitated double helicoids formation.

### Screw dislocations analysis of homochiral helicoids

It was revealed that the formation of hierarchically self-assembled homochiral helicoids was associated with the growth of chiral subunits driven by screw dislocations. Taking R-PMDI-Δ/Pyr helicoids formed in DMF/H$_2$O as an example, as shown in Fig. 2f, the CT assemblies of R-PMDI-Δ and Pyr firstly grew into a curved microsheet as the subunit due to M-helical growth around a screw dislocation (Figs. 2f, g and Supplementary Fig. 16). SEM images showed that the arc-shaped curved subunits arranged in a helical pattern around a center by each upper layer twisting and gathering toward the center based on the lower layer (Fig. 2h), which indicated that new screw dislocations sites continuously generated on the surface of the lower layer and successively created new layers. This self-perpetuating growth step advanced the macroscopic twist level of the co-assemblies and resulted in a twist angle θ between adjacent layers (Fig. 2h). We measured the twist angle θ and counted the layer number. An approximately linear relationship between the twist angle θ and the layer number was found and the linear fittings showed an average twist of 16.9° per layer (Fig. 2i), which proved that the growth evolution mechanism of helicoids conformed to the screw dislocations model[32,41]. Atomic force microscope (AFM) was performed to provide height information of helicoids and further insights into the possible assembly mechanism (Figs. 2j, k, Supplementary Figs. 17 and 18). The AFM images revealed the 3D morphology with a height at the micrometer level, helical arrangement and left-handed helicity (M-helicity) of R-Δ/Pyr helicoids$^{DMF}$ (Supplementary Fig. 17). The characterization of the edge height of the helicoids$^{DMF}$ showed the same screw dislocation features as observed in the SEM images, that is, the nanosheets with a thickness of ca. 41-61 nm continuously generated twisted stacking toward the center (Figs. 2j and k). The AFM images of the R-Δ/Pyr helicoids$^{NMP}$ also exhibited similar results (Supplementary Fig. 18).

The SEM was used to monitor the evolution of helicoids$^{DMF}$ by separating precipitates at different stages of assembly. The sample without annealing after anti-solvent assembly showed a hexagonal tubular structure without macroscopic chirality (Fig. 2l and Supplementary Fig. 19). After thermal annealing, nanosheets smaller than 1 μm were formed as the substrates for the growth of chiral microsheet subunits. It could be found after natural cooling for 2 minutes, some circular microsheets with arc-shaped curved subunits at the edge were produced. The size of the assemblies increased with the prolonging of cooling time. After cooling for 10 minutes, the size of the assemblies reached about 10 μm. It was worth noting that for these samples with a cooling time of 5–10 minutes, a slightly twisted helicoid$^{DMF}$ with macroscopic left-handedness was observed and similar structures could also be observed in helicoids$^{NMP}$ (Supplementary Fig. 20). These structures displayed cyclic subunits on the edges of the nearly circular bottom. These features were consistent with the growth model driven by screw dislocations[29,37,44,53]. After natural cooling for 30 minutes, a fully twisted left-handed helicoid was formed (Fig. 2l), while further elongation of the time did not cause further growth of the helicoids.

### Single-crystal structures

To gain deeper insights into the CT interaction between PMDI-Δ and Pyr and the origin of the macroscopic chiral morphology, we first cultivated a single crystal of R-PMDI-Δ (Supplementary Fig. 21 and Supplementary Table. 3) and a cocrystal of R-PMDI-Δ and Pyr (Supplementary Fig. 22 and Supplementary Table. 4) in DMF/H$_2$O mixed solvents by a heating-cooling method similar to self-assembly. The R-PMDI-Δ single crystal showed a triclinic structure with a P1 feature of the chiral space group. The adjacent molecules displayed π-π stacking and multiple C-H⋯O interactions with the distance of 3.21-3.34 Å and 2.53-2.69 Å, respectively (Fig. 3b). A two-dimensional (2D) supramolecular pattern was formed in the a-b plane (Fig. 3c), which was stacked into a layered structure along the c-axis (Supplementary Fig. 23). It was worth noting that although R-PMDI-Δ was a chiral molecule, no twisted helical structures were formed in its crystal due to the close face-to-face arrangement of the two adjacent molecules (Fig. 3a), which could be responsible for the formation of achiral nanostructures observed in PMDI-Δ assemblies.

Single-crystal X-ray analysis indicated that R-PMDI-Δ/Pyr cocrystal$^{DMF}$ adopted the monoclinic chiral space group C2 with an acceptor to donor ratio of 2:1. Interestingly, two electron acceptors R-PMDI-Δ with symmetry equivalence and one electron donor Pyr formed a twisted acceptor-donor-acceptor structure (ADA sandwich) through π-π stacking and C-H⋯π interaction, where the distance of π-π stacking ranged from 3.19 to 3.35 Å and that of C-H⋯π interaction was 2.74 Å (Fig. 3d). The dihedral angle α between two PMDI-Δ molecule planes in an ADA sandwich was 46.6° or 42.6° (Supplementary Fig. 24). The same ADA sandwiches form M-helical 1D supramolecular co-assemblies through C-H⋯π interactions between R-PMDI-Δ in the way of vertex-to-edge (the distance of C-H⋯π is 2.81 Å) (Figs. 3e and g). This result indicated that R-PMDI-Δ/Pyr cocrystal$^{DMF}$ showed chirality at the supramolecular level, which provided a prerequisite for the formation of macroscopic chiral shapes of co-assemblies[54]. The 1D supramolecular co-assemblies formed 2D supramolecular tiling structures in the b-c plane through C-H⋯O interactions between R-PMDI-Δ (The distance of C-H⋯O was 2.45-2.72 Å) (Figs. 3f and h), which further stacked into layered structures along the a-axis (Fig. 3i).

Fortunately, by slowly evaporating water into the NMP solution mixed with R-PMDI-Δ and Pyr, we also obtained the R-PMDI-Δ/Pyr cocrystal$^{NMP}$ (Supplementary Fig. 25 and Supplementary Table. 5), which displayed a similar composition and structure to cocrystal$^{DMF}$. For example, it also crystallized in the monoclinic C2 chiral space group, and also formed ADA sandwiches (Fig. 3j), M-helical CT co-assemblies (Fig. 3k) and supramolecular tiling structures (Fig. 3l). However, the dihedral angle α of cocrystal$^{NMP}$ was 45.8° and 41.5°, which was slightly smaller than that of cocrystal$^{DMF}$ (Supplementary Fig. 24). In addition, the crystal structures showed that solvent DMF or NMP formed hydrogen bonds with H$_2$O in the intrinsic pore of the R-PMDI-Δ (Supplementary Fig. 26). This result indicated that the hydrogen bonds played an important role in supramolecular co-assembly in the present systems.

### Chiroptical properties of homochiral helicoids

The absorption spectra of PMDI-Δ, Pyr and co-assemblies were shown in Fig. 4a. The helicoids$^{DMF}$ and helicoids$^{NMP}$ displayed a wide absorption band ranging approximately from 383-500 nm and 381-510 nm, respectively, which confirmed the intermolecular CT interaction between PMDI-Δ and Pyr and the formation of the CT complexes. The optical bandgaps for helicoids could be calculated from the edge of the absorption spectra. The results revealed that the bandgap for helicoids$^{DMF}$ was 2.50 eV and that for helicoids$^{NMP}$ was 2.46 eV. Density functional theory (DFT) calculations for the frontier molecular orbitals of CT-complexes of PMDI-Δ and Pyr showed that the highest occupied molecular orbitals (HOMO) of CT-complexes were mainly concentrated on electron donor Pyr and the lowest unoccupied molecular

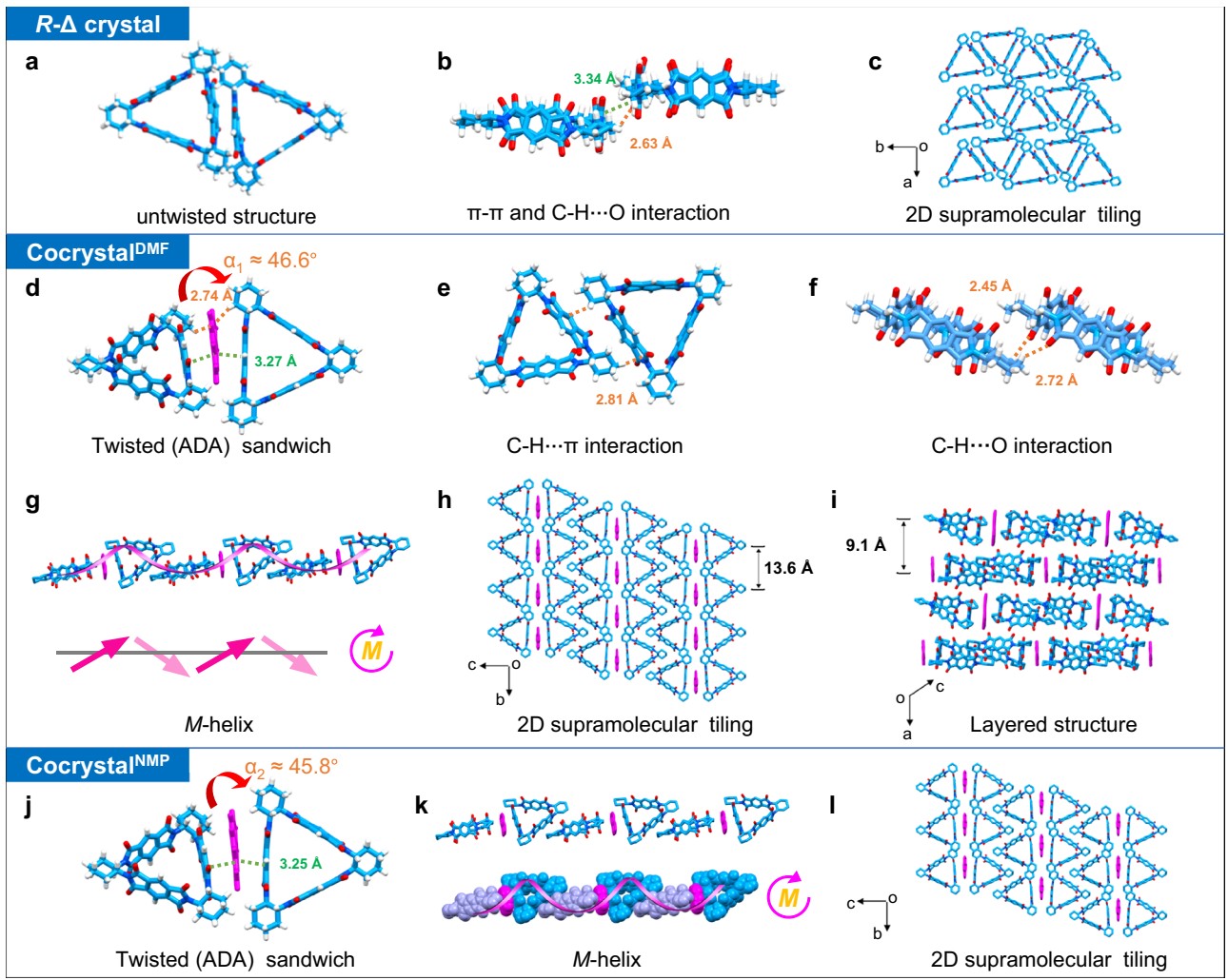

**Fig. 3 | Single-crystal structures. a–c** Single crystal structures of *R*-PMDI-Δ.
**a, b** The untwisted structure formed by π-π stacking and C-H···O interactions in two adjacent molecules. **c** The 2D supramolecular structure formed by *R*-PMDI-Δ in the *a-b* plane. **d–i** The structure of *R*-PMDI-Δ/Pyr cocrystal$^{DMF}$. **d** The π-π stacking and C-H···π interactions in an ADA sandwich. **e** The C-H···π interactions and (**f**) the C-H···O interactions in two adjacent ADA sandwiches. **g** Crystal structure illustrating adjacent ADA sandwiches showed supramolecular *M*-helix by C-H···π interaction. **h** 2D supramolecular tiling formed by ADA sandwichs in the *b-c* plane. **i** Layered stacking structure along the *a*-axis. **j–l** The *R*-PMDI-Δ/Pyr cocrystal$^{NMP}$ showed a similar structure to cocrystal$^{DMF}$, including the ADA sandwichs (**j**) with supramolecular *M*-helix (**k**) and 2D supramolecular structure (**l**). For clarity, solvent molecules are omitted. Except for the panels (**a, b, d, e, f** and **j**), H atoms are omitted in other panels.

orbitals (LUMO) were mainly located on the electron-deficient PMDI-Δ (Supplementary Fig. 27). The calculated HOMO-LUMO energy gaps of CT-complex$^{DMF}$ and CT-complex$^{NMP}$ were 2.63 eV and 2.67 eV, respectively, which were consistent with their optical bandgaps. Distinct from individual Pyr or PMDI-Δ assemblies, the co-assemblies exhibited visible bright green fluorescence under 365 nm UV light (Supplementary Figs. 3b-c). The fluorescence spectra showed Pyr displayed monomer emission peaks at 368 nm and 397 nm and PMDI-Δ showed monomer emission peaks at 413 nm in DMF solution (Fig. 4d), while the co-assemblies clearly showed a red-shifted CT emission peak. The maximum emission of helicoids$^{DMF}$ was 533 nm and that of helicoids$^{NMP}$ was 539 nm, which red-shifted by 120 nm and 126 nm relative to the PMDI-Δ monomer, respectively (Fig. 4d).

The *R*-PMDI-Δ/Pyr helicoids$^{DMF}$ showed a positive CD signal over 400 nm corresponding to the CT absorption band of helicoids$^{DMF}$. A mirrored CD signal was found for the *S*-PMDI-Δ/Pyr helicoids$^{DMF}$ (Fig. 4b and Supplementary Fig. 28). Compared with the PMDI-Δ monomer (Supplementary Fig. 29), the CD signal displayed an obvious red shift, which indicated that the molecular chirality of PMDI-Δ had

been transferred to CT co-assemblies. Similar but stronger CD signals (Fig. 4c and Supplementary Fig. 28) as well as higher $|g_{abs}|$ values were recorded for helicoids$^{NMP}$ (Supplementary Fig. 30). Circularly polarized luminescence (CPL) was carried out to study the excited state chirality of co-assemblies. The CPL signal of PMDI-Δ self-assemblies was not detected due to their weak luminescent properties (Supplementary Fig. 31). However, the PMDI-Δ/Pyr co-assemblies not only expressed macroscopic helical chirality but also exhibited excellent CPL properties. The intense and mirrored CPL signals were detected in both co-assemblies (Figs. 4e, f and Supplementary Fig. 32). As an important parameter to evaluate CPL, the dissymmetry factor $|g_{lum}|$ was found to reach 0.050 for helicoids$^{NMP}$ (Supplementary Fig. 33). The helicoids$^{DMF}$ also showed a high $|g_{lum}|$ value of 0.026 (Supplementary Fig. 33). Furthermore, the CT co-assemblies without macroscopic chirality prepared by annealing in either THF/H$_2$O or dioxane/H$_2$O did not exhibit CT-CPL signals (Supplementary Fig. 34), indicating a strong correlation between chiral helical structures and chiroptical properties. The photophysical properties of co-assemblies were summarized and shown in Supplementary Table 6.

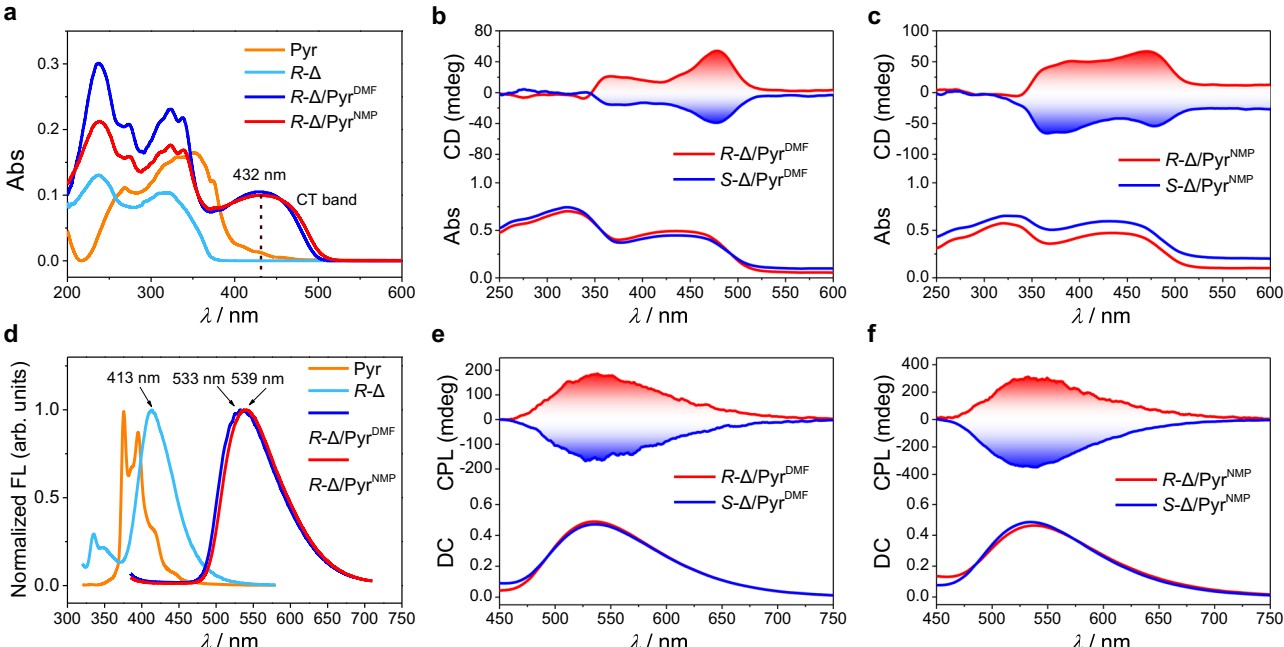

**Fig. 4 | Chiroptical properties of homochiral helicoids. a** UV–Vis absorption spectra of Pyr powder, *R*-PMDI-Δ self-assemblies and PMDI-Δ/Pyr co-assemblies. **b** and **c** CD spectra of co-assemblies. **d** Normalized fluorescence spectra of Pyr monomer, *R*-PMDI-Δ monomer and co-assemblies. [Pyrene] = [*R*-PMDI-Δ] = 0.1 mM, the solvent is DMF. **e**, **f** CPL spectra of co-assemblies. The $\lambda_{ex}$ used for fluorescence and CPL measurements of co-assemblies is 365 nm. The $\lambda_{ex}$ used for fluorescence measurements of Pyr monomer and *R*-PMDI-Δ monomer is 300 nm.

## Self-assembly mechanism of homochiral helicoids

It should be noted that a series of experimental results had shown that the composition and structure of helicoids obtained after annealing were consistent with the corresponding cocrystals except for the emergence of unique screw dislocations in the helicoids. Firstly, the [1]H-NMR spectra showed that the molar ratio of PMDI-Δ and Pyr in the helicoids was very close to 2:1 (Supplementary Fig. 35 and Supplementary Table 7), whether in DMF/H$_2$O or NMP/H$_2$O. This was in agreement with the ratio of the two molecules in the cocrystals. Moreover, the powder X-ray diffraction (PXRD) patterns of the helicoids were consistent with simulated XRD patterns of the corresponding cocrystal (Supplementary Figs. 36 to 39). In addition, we analyzed the PXRD patterns of the samples before and after annealing. Several extra diffraction peaks were presented in the PXRD pattern of the samples before annealing, indicating the existence of some residual self-assemblies of PMDI-Δ and not well-ordered structures. After annealing, these peaks disappeared and completely co-assembled complexes with ordered stacking structures were formed (Supplementary Figs. 40 and 41).

Based on the systematic study of morphology, crystallography and spectroscopy, a possible assembly mechanism was proposed. *R*-(or *S*-) PMDI-Δ and Pyr could co-assemble into ADA sandwichs with helical chirality at the supramolecular level by size matching π-π stacking and intermolecular CT interactions. During the cooling process after annealing, the chiral defects of screw dislocations were generated, which led to the formation of homochiral 3D helicoids by continuously producing new chiral microsheets as subunits.

## Discussion

In summary, we proposed a screw dislocation-driven co-self-assembly strategy to prepare homochiral 3D organic CPL-active helicoids, which was achieved by solution-processable supramolecular co-assembly through thermal annealing. By exploring the chiral expression at different length scales, a continuous chiral transfer mechanism from molecular, to supramolecular, and eventually to macroscopic levels was elucidated. Crystallographic data revealed the transfer of PMDI-Δ molecular chirality to supramolecular chirality of ADA sandwiches,

which was accomplished by sandwiching Pyr between chiral PMDI-Δ molecules. Additionally, morphology observation further demonstrated that the construction of homochiral helicoids with ten micrometers-sized scales was realized in a hierarchical self-assembly driven by screw dislocations. In these processes, the chirality was strictly governed by the molecular chirality of the PMDI-Δ at different length scales. Furthermore, spectroscopic results showed that these helicoids could emit strong circularly polarized luminescence with high |$g_{lum}$| values. Our work not only presented an vital example of 3D organic helicoids with macroscopic chirality in supramolecular systems, but also provided a reliable and simple method for the controlled fabrication of helicoid assemblies with specific functions.

## Methods
### Materials

All chemicals and solvents were used without further purification. Pyromellitic dianhydride (>98.0%) was purchased from Energy-Chemical. Pyrene (>98.0%) was purchased from J&K Scientific. (1 *R*,2 *R*)-(-)−1,2-cyclohexanediamine (>98.0%, TCI), (1 *S*,2 *S*)-(+)−1,2-cyclohexanediamine (>98.0%, TCI), *N*,*N*-dimethylformamide (DMF, Concord Technology (Tianjin) Co., Ltd). AR, >99.0%), 1-Methyl-2-pyrrolidone (NMP, Inno-chem, GC, >99.5%), deionized water (Milli-Q water, 18.2 MΩ·cm) was prepared in the laboratory. The synthetic procedures of (*RRRRRR*)-PMDI-Δ (*R*-Δ) and (*SSSSSS*)-PMDI-Δ (*S*-Δ) were listed in Supplementary Fig. 1. The *rac*-PMDI-Δ was prepared by mixing *R*-PMDI-Δ and *S*-PMDI-Δ with equal mass.

### Co-assembly protocol in DMF/H$_2$O

Typically, *R*/*S*-PMDI-Δ (2.00 mg, 2.25 μmol, 1 eq) and pyrene (0.68 mg, 3.36 μmol, l.5 eq) were dissolved in 550 μL DMF by ultrasound to form a transparent solution. Then, 450 μL H$_2$O (anti-solvent) was added, resulting in a yellow-green suspension (total volume 1000 μL, DMF/H$_2$O $v/v$ = 55%/45%, [*R*/*S*-PMDI-Δ] = 2.25 mM, [Pyrene] = 3.36 mM). The suspension was then annealed at 140 °C for 10 minutes until the charge-transfer (CT) complex between the two molecules was broken and the yellow color of the mixture completely faded. After natural

cooling to room temperature, the white suspension was expected to turn yellow-green again. The precipitates were then separated by centrifugation (13000 r/min, 12000×$g$) and washed twice with deionized water to remove any residual organic solvent. Finally, transfer the precipitates to a clean silicon wafer, glass slide or quartz cuvette for further characterization. The detailed assembly process was shown in Supplementary Figs. 2 and 3.

To gain more insights into morphology evolution and assembly mechanism, it is essential to capture the initial structure of assemblies. Hence, it is requisite to promptly transfer the samples after annealing to a silicon wafer and remove any remaining organic solvent. For the preparation of the samples presented in Fig. 2l, which depicted a natural cooling time of 0 and 2 minutes respectively, a precipitation separation method distinct from centrifugation was employed. That is, for the samples that had just been annealed and the samples that had been left at room temperature for 2 minutes after annealing, 50 μL suspensions were pipetted and rapidly dropped onto a silicon wafer. Following that, the upper layer of liquid was removed, and the remaining precipitates on the silicon wafer were washed twice with 50 μL of deionized water. Finally, the samples were left to dry naturally at room temperature and were characterized by SEM after being coated with a thin layer of Pt in the same way as other samples.

### Co-assembly protocol in NMP/H$_2$O
Typically, $R/S$-PMDI-Δ (2.00 mg, 2.25 μmol, 1 eq) and pyrene (1.14 mg, 5.64 μmol, 2.5 eq) were dissolved in 600 μL NMP by ultrasound to give a transparent solution. Subsequently, 400 μL H$_2$O (anti-solvent) was added and a yellow-green suspension was obtained (total volume 1000 μL, NMP/H$_2$O $v/v$ = 60%/40%, [$R/S$-PMDI-Δ] = 2.25 mM, [Pyrene]= 5.63 mM). The subsequent processing method of the suspension was the same as the protocol in DMF/H$_2$O.

### Self-assembly protocol
The self-assembly protocol of PMDI-Δ was the same as the co-assembly protocol above, except that the electron donor pyrene was not added.

### NMR and mass spectra
$^1$H NMR and $^{13}$C NMR spectra were recorded on a Bruker Advance spectrometer ($^1$H: 400 MHz, $^{13}$C: 100 MHz) in CDCl$_3$ at 298 K. Mass spectra were obtained on Bruker Solarix mass spectrometer (for ESI). The NMR and mass spectra of PMDI-Δ were shown in Supplementary Figs. 42 to 47.

### Scanning electron microscopy (SEM), atomic force microscopy (AFM) and transmission electron microscopy (TEM)
SEM was performed on S4800 (Hitachi, Japan) with an accelerating voltage of 10 kV and a working current of 10 μA to ex situ characterize the co-assemblies. Before SEM measurements, the samples on silicon wafers were coated with a thin layer of Pt to increase the contrast. AFM images were obtained on a Dimension FastScan (Bruker), using ScanAsyst mode under ambient conditions. Fastscan B probes were used for the scan, and samples were prepared by dropping the aqueous dispersion of the co-assemblies onto a double-sided tape. TEM images were operated on a JEOL-2100F electron microscope operating at accelerating voltages of 200 kV.

### Fluorescence microscope image (FM)
The co-assemblies were observed on an IX 83 (Olympus) fluorescence microscope. The samples were prepared by dropping the aqueous dispersion of the co-assemblies onto the glass slide and naturally dried solvent.

### Ultraviolet and visible absorption spectroscopy (UV-Vis)
UV-Vis spectra were measured on a UV-Vis spectrometer (UV-2600 Shimadzu). Solution samples were loaded in a quartz cuvette with

2 mm optical path for pyrene and 1 cm optical path for PMDI-Δ. Solid samples were uniformly dispersed into BaSO$_4$ powders to record the UV-Vis diffuse-reflectance spectra. The optical bandgaps ($E_o$) for helicoids were calculated from the maximum edge ($\lambda_{max}$) of the absorption spectrum, given by the formula $E_o = 1240 / \lambda_{max}$.

### Fluorescence spectra (FL)
The fluorescence spectra were recorded on a F-4600 fluorescence spectrophotometer (Hitachi) at a voltage of 400 V with a 5 nm slit for both the excitation and emission sides. Solution samples were loaded in a quartz cuvette with 2 mm optical path for pyrene and 1 cm optical path for PMDI-Δ. The aqueous dispersion of the solid samples was charged in 1 mm quartz cuvette for FL spectra measurement. Fluorescence decay curves were recorded on a FLS 980 (Edinburgh Instruments). The wavelength of the excitation laser was 358.4 nm. Fluorescence quantum yields were measured on a FluoroMax+ (HORIBA) instrument by using an integrating sphere.

### Circular dichroism (CD) and linear dichroism (LD) spectrum
Solution samples were loaded in a quartz cuvette with 2 mm optical path for pyrene and 1 cm optical path for PMDI-Δ. They were recorded on CD spectrometer J-815 (JASCO) at a scanning rate of 500 nm min$^{-1}$ in the range of 200 - 600 nm. The CD and LD spectra of solid samples were simultaneously recorded for each sample on CD spectrometer J-1500 (JASCO) under a diffuse-reflectance mode at a scanning rate of 500 nm min$^{-1}$ in the range of 200 - 800 nm. CD and LD spectra of each solid sample were recorded more than 3 times to ensure the chiroptical signals. The absorptive dissymmetry factor ($g_{abs}$, also known as $g_{CD}$) spectra were directly transferred from CD spectra using the SpectraManager software of JASCO. The $g_{abs}$ can be used to quantify the magnitude of CD, given by the formula $g_{abs} = 2 (\varepsilon_L - \varepsilon_R)/(\varepsilon_L + \varepsilon_R)$, where $\varepsilon_L$ and $\varepsilon_R$ refer to the extinction coefficients for left- and right-handed circularly polarized light, respectively.

### Circularly polarized luminescence spectrum (CPL)
The CPL spectra were recorded on CPL-300 spectrophotometer (JASCO) in a range of 400 - 750 nm. The aqueous dispersion of the solid samples was charged in 1 mm quartz cuvette for CPL spectra measurement. The slits for both the excitation and emission sides were 3000 μm. The luminescence dissymmetry factor ($g_{lum}$) spectra were transferred from CPL spectra using the SpectraManager software of JASCO. The $g_{lum}$ was used to quantify the extent of chiral fluorescence dissymmetry, given by the formula $g_{lum} = 2 (I_L - I_R)/(I_L + I_R)$, where $I_L$ and $I_R$ represent the intensities of left and right circularly polarized light, respectively. CPL spectra of each solid sample were recorded more than 3 times by flipping and rotating the cuvette to ensure the chiroptical signals.

### Powder X-ray diffraction (XRD) measurements
The solid samples were loaded directly onto a glass sample holder to record the X-ray diffraction spectra on EmpyreanX (PANalytical B.V.) with Cu/Kα radiation ($\lambda = 1.5406$ Å) at 40 kV and 40 mA. The scanning range was from 1° to 40°.

### Single crystal X-ray diffraction (XRD) measurements
The single-crystal XRD was measured by XtaLAB Synergy-R (Rigaku). The structures were solved by direction methods and refined by a full matrix least squares technique based on F2 using SHELXL 97 program (Sheldrick, 1997).

### $R$-PMDI-Δ single crystal$^{DMF}$
In a 2 mL vial, 1.50 mg $R$-PMDI-Δ was dissolved in 400 μL DMF by heating and then 200 μL deionized water was added. Some white precipitates were formed immediately. After sealing the vial, the precipitates were fully dissolved by heating. Colorless transparent plate

crystals were observed after standing at room temperature for about 2 days. Pick the right crystal for X-ray single crystal test.

### R-PMDI-Δ/Pyr cocrystal$^{DMF}$

In a 2 mL vial, 0.75 mg R-PMDI-Δ and 0.17 mg pyrene were dissolved in 266 μL DMF by heating. Following this, 133 μL deionized water was added, resulting in the formation of yellow precipitates. The vial was then sealed and the precipitates were fully dissolved by heating. Pale yellow flake R-Δ/Pyr cocrystals$^{DMF}$ were prepared after standing at room temperature for about 2 days. Pick the right crystal for X-ray single crystal test.

### R-PMDI-Δ/Pyr cocrystal$^{NMP}$

1.0 mg R-PMDI-Δ and 1.82 mg pyrene were dissolved in 600 μL NMP by heating. The solution was then cooled and filtered with a 0.22 μm microporous membrane. The filtrate was transferred to a 2 mL vial which was placed in a 20 mL vial containing 3 mL of water. With slow vapor diffusion of H$_2$O into the NMP solution of R-PMDI-Δ and pyrene, high-quality yellow flake R-Δ/Pyr cocrystals$^{NMP}$ formed after three days. Pick the right crystal for X-ray single crystal test.

### Density functional theory calculation

Density functional theory (DFT) computation was performed by Gaussian 09 Revision D.01 program at B3LYP 6-311 G** level. The initial structures of R-PMDI-Δ, CT-complexes and pyrene were extracted from corresponding single crystal data, respectively, and without further geometries optimization.

## Data availability

All relevant data are available from the corresponding author. Supplementary Information is available in the online version of the paper. The X-ray crystallographic coordinates for structures reported in this study have been deposited at the Cambridge Crystallographic Data Centre, under deposition number CCDC: 2322342 (R-PMDI-Δ crystal$^{DMF}$), 2322343 (R-Δ/Pyr cocrystal$^{DMF}$) and 2322344 (R-Δ/Pyr cocrystal$^{NMP}$). Source data are provided with this paper.

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

## Acknowledgements

We thank X.F. Zhu of the Lanzhou University and G.H. Ouyang of the Institute of Chemistry, Chinese Academy of Sciences (ICCAS) for their insightful discussions, and T.L. Liang of the ICCAS for the assistance in solving and refining single crystal structures. We also express our gratitude to H.X. Wang of the ICCAS for her significant help in improving the language of the paper. This work was supported by the National Natural Science Foundation of China (21890734, 21890730, and 22202209).

## Author contributions

M.L. conceived the idea and supervised the project. S.W. and C.D. carried out the molecular synthesis. S.W. performed the assembly experiments and DFT computations. S.W. and X.S. analyzed the data. S.W. and M.L. wrote the manuscript.

## Competing interests

The authors declare no competing interests.
