## [Peer Review File · Nature Communications]

Macroscopic Homochiral Helicoids Self-Assembled via Screw DislocationsREVIEWER COMMENTS

Reviewer #1 (Remarks to the Author):

In this work, Liu et al. constructed the macroscopic twisted three-dimensional (3D) homochiral structures with ten micrometers-sized scales through the screw dislocation-driven self-assembly strategy. These microstructures exhibit distinct macroscopic homochiral helicity governed by molecular chirality. The continuous adjustment of the chiral morphology by the addition ratio of the donor was also realized. The chiral structures exhibit mirror-symmetric CD signals and strong CPL signals. The authors used single-crystal X-ray diffraction to study their self-assembly processes, and a self-assembly mechanism driven by screw dislocation is proposed. Rigid macrocyclic molecules are prone to form planar organic assemblies/crystals. Here, when co-assembly with other aromatics, chiral twisted assemblies or crystals which display non-Euclidean geometry with curved surfaces via a large scale chirality transmission can be formed. This breaks the convention understanding of macrocyclic supramolecular assembly. In my opinion, this research is a breakthrough in supramolecular chemistry/self-assembly/chiral science/topology. The manuscript is well-written and clear. I recommend its publication after following minor revisions.

1. In the section "Introduction", the authors should add correlative references about "non-Euclidean geometry" and "macroscopic chirality" to reinforce the statement. Although screw dislocation-driven chiral morphologies are typically observed in inorganic, organic-inorganic hybrids, and polymers, constructing 3D helical homochiral based on organic molecules through screw dislocation has not been reported. The authors should address this point.
2. In this paper, two solvents (tetrahydrofuran and dioxane) were also tried. Microsheets were obtained in both solvents, but no chiral structure was observed. The authors should supplement the CPL information of those microsheet structures, to prove whether the macroscopic twisted chiral morphology has stronger chiral optical activity than ordinary microsheets.
3. In addition to DMF and NMP mentioned in the article, can this twisted chiral structure be formed in other solvents?
4. In NMP, the authors observed an unusual S-shaped double helicoid structure. I think this structure is very interesting because it may involve a higher level of self-assembly. Therefore, the author should discuss the relationship between the two chiral structures and the formation mechanism of the double helicoids.
5. There are still some mistakes that need to be corrected.
 - (1) The corresponding figure in the chapter on the single-crystal structure should be Fig. 3 instead of Fig. 4. (lines 125, 128, 132, 137 and 138)
 - (2) In line 186, the word "because" should be removed;
 - (3) The ¹H-NMR spectrum in Fig. S25 should show the MHz number of the instrument.

(4) In line 253 of Methods and in the penultimate line of Fig. S9, "is" should be replaced with "were".

Reviewer #2 (Remarks to the Author):

The report by Liu and coworkers details the co-assembly of chiral PMDI macrocycles with pyrene via intermolecular charge transfer complexes to form screw dislocated helicoid structures with efficient transfer of chirality. The CT complex results in intense emission of circularly polarized luminescence (CPL) with $g(\text{lum})$ reaching 0.05 which is quite high for colloidal dispersions. Overall, the work is very thorough with both the chiroptical, diffraction and morphological analysis well supporting the claims of the manuscript. On top of this, the field of chiral materials is very hot these days and this offers a nice and novel approach to small molecule to supramolecular chirality transfer. This work should be well suited for publication in Nature Comm after the following revisions are taking into consideration.

1) There are several grammatical and tense issues throughout the manuscript that should be corrected prior to publication.

2) In the abstract, the authors state that this is the first time that organic helicoids were formed from small molecule supramolecular assembly. This seems very presumptive and I do not believe this to be true. This should be removed.

3) Can AFM be performed on these materials to get analogous dimensional information in the Z-direction (i.e., heights of helicoids) for comparison?

Reviewer #3 (Remarks to the Author):

This manuscript reports a comprehensive study on the macroscopic helicoids from PMDI- Δ and pyrene. The approach is based on logical molecular design and analysis based on the solid methodology, particularly x-ray analysis, which provides complementary information and insight into this original system, which is particularly challenging in molecular assembling structures. The results and analysis support the conclusions.

Nevertheless, there are some shortcomings in the discussions which need to be clarified.

The origin of the differences in structures between DMF and NMP is not very clear. How "complex S-shaped double helicoids" are observed with NMP? It would be helpful if the authors provided more quantitative propositions that link the properties of DMF and NMP vs. the structures.

The analysis of the powder diffraction should be more developed; the data shown in S26-28 only show that the peaks are different before and after the annealing. If possible, a more thorough analysis of the origin of the disappearance/appearance of certain peaks can be provided.

Obviously, the tuning of the structures can be the next step. An interesting analysis explains why Pyrene has the matching structure as the electron acceptor. How about the PMDI part? Can one use other PMDI derivatives, for example, with longer linkage or higher stacking moieties, and if so, how may that lead to tuning the assembly structures? Authors may give some thoughts on the question.

Some of the paragraphs are quite heavy to read, and it requires reading several times to get the principal messages. Also, the co-assemblies before and after annealing should be more clearly noted, probably with consistent notations.

some minor comments are shown below:

in the abstract, "both the small enantiomeric PMDI- Δ and pyrene", may be misleading. It may be better to remove "both".

The description of what PMDI is the abbreviation of is needed.

Replies to the Reviewers' Comments

We would like to thank all the Reviewers for their positive comments, insightful suggestions, and valuable opinions on improving the quality of our manuscript. We have thoroughly considered all the comments and made changes as required to fully address them. Please find our replies (blue font) below, which follow the corresponding comments (black font). The details of the changes made in our revised manuscript have been highlighted in yellow in a separate PDF document.

For Reviewer #1:

In this work, Liu et al. constructed the macroscopic twisted three-dimensional (3D) homochiral structures with ten micrometers-sized scales through the screw dislocation-driven self-assembly strategy. These microstructures exhibit distinct macroscopic homochiral helicity governed by molecular chirality. The continuous adjustment of the chiral morphology by the addition ratio of the donor was also realized. The chiral structures exhibit mirror-symmetric CD signals and strong CPL signals. The authors used single-crystal X-ray diffraction to study their self-assembly processes, and a self-assembly mechanism driven by screw dislocation is proposed. Rigid macrocyclic molecules are prone to form planar organic assemblies/crystals. Here, when co-assembly with other aromatics, chiral twisted assemblies or crystals which display non-Euclidean geometry with curved surfaces via a large scale chirality transmission can be formed. This breaks the convention understanding of macrocyclic supramolecular assembly. In my opinion, this research is a breakthrough in supramolecular chemistry/self-assembly/chiral science/topology. The manuscript is well-written and clear. I recommend its publication after following minor revisions.

Reply: We sincerely thank you for your careful reading of the manuscript, profound comments, and the recommendation.

1-1. In the section "Introduction", the authors should add correlative references about "non-Euclidean geometry" and "macroscopic chirality" to reinforce the statement.

Reply: We sincerely appreciate your valuable comment. We have checked the references carefully and added correlative references on "Euclidean geometry", "non-Euclidean surfaces" and "macroscopic chirality" into the introduction part of the revised manuscript.

On "Euclidean geometry" and "non-Euclidean surfaces", Greenberg, M. J. *Euclidean and non-Euclidean geometries: Development and history*. (Macmillan, 1993).

On "macroscopic chirality", Morrow, S. M., Bissette, A. J. & Fletcher, S. P. Transmission of chirality through space and across length scales. *Nat. Nanotechnol.* 12, 410–419 (2017).

1-2. Although screw dislocation-driven chiral morphologies are typically observed in inorganic, organic-inorganic hybrids, and polymers, constructing 3D helical homochiral based on organic molecules through screw dislocation has not been reported. The authors should address this point.

Reply: We appreciate your very important point. A systematic literature review was conducted and your suggestions were supported by the results. We have found that screw dislocations are a classical method for producing 3D spirals/helical structures from inorganic crystals such as metal crystals, transition-metal dichalcogenides and polymers such as polyethylene glycol and conjugated diblock copolymer. However, it is rare to obtain uniform and well-structured 3D spirals/helical structures and to precisely control their scales and shapes from small organic molecules using screw dislocations.

To address your suggestion and emphasize this point, we have added the following description to the introduction part of the revised manuscript (Page 2, line 34). *“In addition, screw dislocation is also a classical method for producing three-dimensional (3D) spirals from inorganic crystals and polymers. However, 3D helicoid structures rarely originate from small organic molecules.”* A related reference to this statement was also added.

2. In this paper, two solvents (tetrahydrofuran and dioxane) were also tried. Microsheets were obtained in both solvents, but no chiral structure was observed. The authors should supplement the CPL information of those microsheet structures, to prove whether the macroscopic twisted chiral morphology has stronger chiral optical activity than ordinary microsheets.

Reply: The CT assemblies obtained in THF/H₂O and Dioxane/H₂O showed a general microsheet morphology without macroscopic chirality. We tested the CPL spectra of these two coassemblies, and the results were presented in Fig. R1 (Fig. S30 in the revised version). Both of these CT complexes failed to show CPL signals. This suggested that only helicoids with macroscopic chirality can generate CPL signals, highlighting the uniqueness of the helicoid morphology.

In the revised manuscript, we have added a description to the “Chiroptical properties of homochiral helicoids” section (Page 8, line 191) to strengthen the comparison, that is *“Furthermore, the CT coassemblies without macroscopic chirality prepared by annealing in either THF/H₂O or dioxane/H₂O did not exhibit CT-CPL signals (Fig. S30), indicating a strong correlation between chiral helical structures and chiral optical properties.”*

Fig. R1. CPL spectra of PMDI- Δ /Pyrr co-assemblies obtained by annealing in THF/H₂O and Dioxane/H₂O, respectively.

3. In addition to DMF and NMP mentioned in the article, can this twisted chiral structure be formed in other solvents?

Reply: Thanks for your good question. In addition to the solvents mentioned in the original version, such as DMF, NMP, THF, and dioxane. Several other solvents that cannot form hydrogen bonds have also been employed to characterize their morphology. The results were shown in Fig. R2, indicating that no helicoid structures could be observed in ethyl acetate/n-hexane, chloroform/n-hexane and acetonitrile/H₂O. These additional data further supported our previous statement, *“This result indicated that the hydrogen bonds play an important role in supramolecular co-assembly in the present systems”*. (in Page 7, line 160)

Fig. R2. SEM images of *R*-PMDDI- Δ /Pyr co-assemblies obtained in different mixed solvents. (a) ethyl acetate/*n*-hexane. (b) chloroform/*n*-hexane. (c) acetonitrile/H₂O. (*R*-PMDDI- Δ 1.0 mg, the mole ratio of *R*-PMDDI- Δ /Pyr is 1 to 3 and the solvent volume ratio is 200 μ L to 200 μ L)

4. In NMP, the authors observed an unusual S-shaped double helicoid structure. I think this structure is very interesting because it may involve a higher level of self-assembly. Therefore, the author should discuss the relationship between the two chiral structures and the formation mechanism of the double helicoids.

Reply: We appreciate you for providing this important suggestion. We conducted a systematic morphological observation to investigate the relationship between the two chiral structures. We have successfully captured some microstructures between helicoids and double helicoids in DMF/H₂O (Fig. R3, Fig. S13 in the revised version). These structures are formed by two helicoids side by side, resembling a double helicoid, but they can not be completely connected and grow to form a whole structure. However, in NMP/H₂O, we found that some complete double helicoids can be observed, and their morphology displays a fully formed structure. (Fig. R4, Fig. S12 in the revised version).

Combined with our initial morphology of coassemblies, we have proposed a possible formation mechanism for helicoids and double helicoids. As shown in Fig. R5 (Figs. 2e and S14 in the new version). Taking *R*-PMDDI- Δ /Pyr^{NMP} as an example, we found that chiral subunits usually tend to stack layer-by-layer, which leads to the formation of helicoids. However, some chiral microsheet subunits can be connected side-by-side while layer-by-layer stacking, which leads to the growth of co-assemblies in two directions, thus forming a double helicoid structure with two helicoids connecting side-by-side.

It is worth noting that the handedness of double helicoids is consistent with that of helicoids.

Accordingly, we have condensed the above description and added the description to the “Self-assembly of Homochiral Helicoids” part in the revised manuscript (Page 4, line 84).

The description in the new version is: “....., except that a fully-formed *S*-type double helicoid (for *R*-PMDDI- Δ /Pyr) and anti-*S*-shaped double helicoid (for *S*-PMDDI- Δ /Pyr) could also be observed in NMP/H₂O (Figs. 2b to 2d and S12). Interestingly, in DMF/H₂O, only two helicoids side by side were observed, but they could not be completely connected and grow to form a whole structure (Fig. S13). By successfully capturing the initial morphology of helicoids and double helicoids in NMP/H₂O, a possible formation mechanism for both types of helicoids was proposed (Figs. 2e and S14). The process involved the layer-by-layer stacking of chiral subunits, which led to the formation of helicoids. However, some chiral microsheets could be connected side-by-side while stacking layer-by-layer, which led to the co-assemblies growing in two directions, thus forming a double-helicoid structure by two helicoids connecting side-by-side.”

Fig. R3. (a-d) SEM images of microstructures between helicoids and double helicoids obtained in DMF/H₂O after annealing. These structures are formed by two helicoids side by side, resembling a double helicoid, but they can not be completely connected and grow to form a whole structure.

Fig. R4. SEM images of a fully-formed S-type double helicoid (for R -PMDI- Δ /Pyr) (a) and a fully-formed anti-S-shaped double helicoid (for S -PMDI- Δ /Pyr) (b) obtained in NMP/H₂O after annealing.

Fig. R5. Schematic diagram of the formation mechanism of (a) helicoids and (b) double helicoids. The SEM images of helicoids and double helicoids in this figure were obtained by annealing assembly in NMP/H₂O mixed solvent.

5. There are still some mistakes that need to be corrected.

(1) The corresponding figure in the chapter on the single-crystal structure should be Fig. 3 instead of Fig. 4. (lines 125, 128, 132, 137 and 138)

(2) In line 186, the word "because" should be removed;

(3) The ¹H-NMR spectrum in Fig. S25 should show the MHz number of the instrument.

(4) In line 253 of Methods and in the penultimate line of Fig. S9, "is" should be replaced with "were".

Reply: Thank you very much for carefully reading. We have carefully checked the manuscript and corrected the errors accordingly. These updates are marked in yellow in the revised edition.

For Reviewer #2:

The report by Liu and coworkers details the co-assembly of chiral PMDI macrocycles with pyrene via intermolecular charge transfer complexes to form screw dislocated helicoid structures with efficient transfer of chirality. The CT complex results in intense emission of circularly polarized luminescence (CPL) with $g(\text{lum})$ reaching 0.05 which is quite high for colloidal dispersions. Overall, the work is very thorough with both the chiroptical, diffraction and morphological analysis well supporting the claims of the manuscript. On top of this, the field of chiral materials is very hot these days and this offers a nice and novel approach to small molecule to supramolecular chirality transfer. This work should be well suited for publication in Nature Comm after the following revisions are taking into consideration.

Reply: We sincerely thank you for your recommendation and your recognition of the importance of our article.

1) There are several grammatical and tense issues throughout the manuscript that should be corrected prior to publication.

Reply: Thank you for your comment. We have carefully reviewed the manuscript and made improvements to address the issue of grammatical and tense. These updates are marked in yellow in the revised edition.

2) In the abstract, the authors state that this is the first time that organic helicoids were formed from small molecule supramolecular assembly. This seems very presumptive and I do not believe this to be true. This should be removed.

Reply: We agree with you and we have removed the statement “the first time” in the revised version of the manuscript. In addition, we have added a qualifier “via screw dislocations” to crystallize our statement. This statement has been revised as “*Our results demonstrate the formation of a homochiral macroscopic organic helicoid and function emergence from small molecules via screw dislocations*”.

3) Can AFM be performed on these materials to get analogous dimensional information in the Z-direction (i.e., heights of helicoids) for comparison?

Reply: Thank you very much for the insightful suggestion. According to your suggestion, we have performed AFM to characterize the heights of helicoids obtained in DMF/H₂O and NMP/H₂O, respectively. The experimental results were shown in Fig. R6 (for *R*- Δ /Pyr helicoid^{DMF}) and R7 (for *R*- Δ /Pyr helicoid^{NMP}). The AFM images clearly revealed the 3D morphology, helical arrangement, and left-handed helicity (*M*-helicity) of helicoid^{DMF}. The height of helicoid^{DMF} reached the micrometer level. Furthermore, by characterizing the chiral subunit nanosheets at the edge of the helicoid^{DMF}, we observed the same screw dislocation features as in the SEM images. The result further indicated that the helicoid was formed by the layer-by-layer dislocation stacking of curved subunits. The measurement results also indicated that the thickness of nanosheets was ca. 41-61 nm. The AFM images of the helicoid^{NMP} also exhibited similar results, with a height of 570 nm and a thickness of nanosheets ca. 45-49 nm.

Accordingly, we have added the following AFM images to the revised manuscripts and supporting information (Figs. 2g, 2k, S16 and S17 in the revised version) and added the corresponding discussion in the “Screw dislocations analysis of homochiral helicoids” part of the revised paper (page 5, line 110).

The description in the new version is: “Atomic force microscope (AFM) was performed to provide height information of helicoids and further insights into the possible assembly mechanism (Figs. 2g, 2k, S16 and S17). The AFM images revealed the 3D morphology with a height at the micrometer level, helical arrangement and left-handed helicity (*M*-helicity) of *R*- Δ /Pyr helicoids^{DMF} (Figs. S16). The characterization of the edge height of the helicoids^{DMF} showed the same screw dislocation features as observed in the SEM images, that is, the nanosheets with a thickness of ca. 41-61 nm continuously generated twisted stacking toward the center (Figs. 2g and 2k). The AFM images of the *R*- Δ /Pyr helicoids^{NMP} also exhibited similar results (Fig. S17).”

Fig. R6. AFM data of *R*- Δ /Pyr helicoid^{DMF} (a) AFM topography image; (b) three-dimensional (3D) AFM topography image. AFM images (a, b) clearly displaying the formation of an *M*-helical microstructure; (c) AFM height profile, as denoted by the blue line in (a), indicating a maximum height of ca. 1.41 μm and a maximum width of ca. 7.21 μm . (d and e) The AFM morphology image and 3D AFM morphology image of the helicoid^{DMF} edge, as denoted by the red box in (a). (f) shows the height profile across the blue line in (e), indicating the thickness of the subunit nanosheets is ca. 41-61 nm.

We have added (e) and (f) in **Fig. R6** above to **Fig. 2** in the revised manuscript, and they correspond to **Fig. 2g** and **Fig. 2k** respectively.

Fig. R7. AFM data of $R-\Delta/\text{Pyr}$ helicoid^{NMP} (a) AFM topography image; (b) three-dimensional (3D) AFM topography image. AFM images (a, b) clearly displaying the formation of an M -helical microstructure; (c) AFM height profile, as denoted by the blue line in (a), indicating a height of ca. 570 nm and a width of ca. 6.95 μm. (d and e) The AFM morphology image and 3D AFM morphology image of the helicoid^{NMP} edge, as denoted by the red box in (a). (f) shows the height profile across the blue line in (d), indicating the thickness of the subunit nanosheets is ca. 45-49 nm.

For Reviewer #3:

This manuscript reports a comprehensive study on the macroscopic helicoids from PMDI- Δ and pyrene. The approach is based on logical molecular design and analysis based on the solid methodology, particularly x-ray analysis, which provides complementary information and insight into this original system, which is particularly challenging in molecular assembling structures. The results and analysis support the conclusions. Nevertheless, there are some shortcomings in the discussions which need to be clarified.

Reply: We appreciate your positive comments. According to your advice, we have made corresponding revisions to improve our manuscript.

(1) The origin of the differences in structures between DMF and NMP is not very clear. How "complex S-shaped double helicoids" are observed with NMP? It would be helpful if the authors provided more quantitative propositions that link the properties of DMF and NMP vs. the structures.

Reply: We thank your constructive suggestions. This comment is similar to Comment #4 for Reviewer #1. According to your advice, we further studied the relationship between the morphology and solvent parameters in DMF/H₂O and NMP/H₂O mixed solvent. The results were as follows.

We have successfully captured some microstructures between helicoids and double helicoids in DMF/H₂O (Fig. R8, Fig. S13 in the revised version). These structures are formed by two helicoids side by side, resembling a double helicoid, but they can not be completely connected and grow to form a whole structure. However, in NMP/H₂O, we found that some complete double helicoids can be observed, and their morphology displays a fully formed structure. (Fig. R9, Fig. S12 in the revised version).

Combined with our initial morphology of coassemblies, we have proposed a possible formation mechanism for helicoids and double helicoids. As shown in Fig. R10 (Figs. 2e and S14 in the new version). Taking *R*-PMDI- Δ /Pyr^{NMP} as an example, we found that chiral subunits usually tend to stack layer-by-layer, which leads to the formation of helicoids. However, some chiral microsheet subunits can be connected side-by-side while layer-by-layer stacking, which leads to the growth of co-assemblies in two directions, thus forming a double helicoid structure with two helicoids connecting side-by-side.

It is worth noting that the handedness of double helicoids is consistent with that of helicoids.

Combined with the above mechanism, we put forward a possible explanation for the difference in morphology between DMF/H₂O and NMP/H₂O. We observed this complete double helicoid morphology in NMP/H₂O mixed solvent and attributed it to the lower solvent polarity and the higher solvent viscosity of NMP compared with DMF (Table R1) (Table S1 in the new version). The polarity parameter E_T^N value of NMP is 0.355, and that of DMF is 0.386⁽¹⁻²⁾. Lower polarity can promote the CT interactions between PMDI- Δ and pyrene. This can be demonstrated by the fluorescence data in the text, where the helicoids obtained in NMP/H₂O show a more red-shifted CT emission than in DMF/H₂O. Secondly, the viscosity of NMP (1.67 mPa·s) is higher than that of the DMF (0.92 mPa·s)⁽²⁻⁴⁾. We speculated that this higher viscosity is conducive to stable dispersion of the chiral subunits in the solution to avoid agglomeration and settling in the early stage of assembly, which provides conditions for side-by-side connecting. The combination of these two factors enables the formation of a fully assembled double helicoid in NMP/H₂O.

Accordingly, we have condensed the above description and added the description to the "Self-assembly of Homochiral Helicoids" part in the revised manuscript (Page 4, line 84).

The description in the new version is: “....., except that a fully-formed S-type double helicoid (for R-PMDI- Δ /Pyr) and anti-S-shaped double helicoid (for S-PMDI- Δ /Pyr) could also be observed in NMP/H₂O (Figs. 2b to 2d and S12). Interestingly, in DMF/H₂O, only two helicoids side by side were observed, but they could not be completely connected and grow to form a whole structure (Fig. S13). By successfully capturing the initial morphology of helicoids and double helicoids in NMP/H₂O, a possible formation mechanism for both types of helicoids was proposed (Figs. 2e and S14). The process involved the layer-by-layer stacking of chiral subunits, which led to the formation of helicoids. However, some chiral microsheets could be connected side-by-side while stacking layer-by-layer, which led to the co-assemblies growing in two directions, thus forming a double-helicoid structure by two helicoids connecting side-by-side. The formation of complete double helicoids in NMP/H₂O could be attributed to the lower polarity and the higher viscosity of NMP compared with DMF (Table. S1). These two factors enhanced intermolecular CT interactions between PMDI- Δ and Pyr and gave rise to a more stable dispersion of chiral subunits in the solution, which enabled the connection of side-by-side and facilitated double helicoids formation.”

Fig. R8. (a-d) SEM images of microstructures between helicoids and double helicoids obtained in DMF/H₂O after annealing. These structures are formed by two helicoids side by side, resembling a double helicoid, but not completely connected and fully assembled to form a whole.

Fig. R9. SEM images of a fully-formed S-type double helicoid (for R -PMDI- Δ /Pyr) (a) and a fully-formed anti-S-shaped double helicoid (for S -PMDI- Δ /Pyr) (b) obtained in NMP/ H_2O after annealing.

Fig. R10. Schematic diagram of the formation mechanism of (a) helicoids and (b) double helicoids. The SEM images of helicoids and double helicoids in this figure were obtained by annealing assembly in NMP/ H_2O mixed solvent.

Table R1.

The viscosity and polarity parameters of solvents (H_2O , DMF and NMP)

Solvent	Viscosity (mPa·s)	Reichardt scale of polarity E_{T}^{N}
H_2O	1.00	1.000
DMF	0.92	0.386
NMP	1.67	0.355

The data in **Table R1** are derived from references (1-4).

(2) The analysis of the powder diffraction should be more developed; the data shown in S26-28 only show that the peaks are different before and after the annealing. If possible, a more thorough analysis of the origin of the disappearance/appearance of certain peaks can be provided.

Reply: Thank you for your valuable suggestions. We carefully analyzed the PXRD patterns of the samples before and after annealing, and the results in DMF/H₂O were shown in Fig. R11 (Fig. S36 in the revised version). Firstly, compared with the PXRD patterns of the samples after annealing, we noticed that the samples before annealing showed some additional diffraction peaks at 1.45 nm, 0.87 nm and 0.73 nm. These diffraction peaks can be attributed to PMDI- Δ self-assemblies by comparing them with the single-crystal simulated XRD patterns (Fig. S34). This suggested that there were still some residual self-assemblies of PMDI- Δ . After annealing, these peaks disappeared, indicating the formation of complete co-assembled complexes. Furthermore, we also observed some diffraction peaks at 1.02 nm and 0.95 nm near 0.91 nm in the samples before annealing. According to the co-crystal structure analysis, the diffraction peak at 0.91 nm corresponded to the interlayer spacing in the layered stacking structure. This indicated that the co-assemblies were not well-ordered before annealing. However, after annealing, these peaks disappear, leaving only a strong diffraction peak at 0.91 nm, which indicates the formation of an ordered layered stacking structure. In addition, as shown in Fig. R12 (Fig. S37 in the revised version), the PXRD patterns of samples before and after annealing in NMP/H₂O were also compared, and similar experimental results can be obtained.

Accordingly, we have condensed the above description and added the description to the “Self-assembly mechanism of homochiral helicoids” part in the revised manuscript (Page 8, line 203).

The description in the new version is: “*In addition, we analyzed the PXRD patterns of the samples before and after annealing. Several extra diffraction peaks were presented in PXRD pattern of the samples before annealing, indicating the existence of some residual self-assemblies of PMDI- Δ and not well-ordered structures. After annealing, these peaks disappeared and completely co-assembled complexes with ordered stacking structures were formed (Figs. S36 and S37).*”

Fig. R11. Comparison of PXRD patterns of the co-assemblies^{DMF} before and after annealing. The figure on the right is an enlarged view of the selected part of the red dotted box in the left figure.

Fig. R12. Comparison of PXRD patterns of the co-assemblies^{NMP} before and after annealing. The figure on the right is an enlarged view of the selected part of the red dotted box in the left figure.

(3) Obviously, the tuning of the structures can be the next step. An interesting analysis explains why Pyrene has the matching structure as the electron acceptor. How about the PMDI part? Can one use other PMDI derivatives, for example, with longer linkage or higher stacking moieties, and if so, how may that lead to tuning the assembly structures? Authors may give some thoughts on the question.

Reply: Thank you for your insightful suggestions. We are currently exploring related triangle systems, but synthesizing and single crystallization is still a challenge. We speculate that by assembling these triangle macrocycle molecules with longer linkers through matching π - π stacking, we can also produce intriguing chiral structures. This is an area of key interest that we will focus on in our next step.

(4) Some of the paragraphs are quite heavy to read, and it requires reading several times to get the principal messages.

Reply: Thank you for your review. We tried our best to improve the manuscript and made changes to the manuscript. These changes will not influence the content and framework of the paper. And here we did not list the changes but marked them in yellow in the revised paper.

(5) Also, the co-assemblies before and after annealing should be more clearly noted, probably with consistent notations.

Reply: Thanks for the suggestion. We have updated the sample information labels in Fig. 1c, Fig. 2l, etc. to make them clearer to show whether annealing was performed. In addition, we added modifiers of "before annealing" or "after annealing" in several positions of the revised manuscript to make them easier to distinguish, such as on "page 3, line 65 and line 66", and on "page 8, line 198 and line 204".

some minor comments are shown below:

(6) in the abstract, "both the small enantiomeric PMDI- Δ and pyrene", may be misleading. It may be better to remove "both".

Reply: We sincerely thank the reviewer for careful reading. As suggested by the reviewer, we have removed the word "both".

(7) The description of what PMDI is the abbreviation of is needed.

Reply: Thanks for your advice. PMDI is short for pyromellitic diimide, and PMDI- Δ is short for pyromellitic diimide-based molecular triangle. We have attached the description of "pyromellitic diimide-based molecular triangle" to PMDI- Δ in the abstract and introduction part.

References

- (1) Germán, L., Cuevas, J. M., Cobos, R., Pérez-Alvarez, L. & Vilas-Vilela, J. L. Green alternative cosolvents to N-methyl-2-pyrrolidone in water polyurethane dispersions. *RSC advances* **11**, 19070-19075 (2021).
- (2) Reichardt, C. Solvatochromic dyes as solvent polarity indicators. *Chemical reviews* **94**, 2319-2358 (1994).
- (3) Bildyukevich, A. *et al.* Effect of the solvent nature on the structure and performance of poly (amide-imide) ultrafiltration membranes. *Journal of Materials Science* **55**, 9638-9654 (2020).
- (4) Gupta, S. Viscometry for liquids. *Cham: Springer International Publishing* (2014).

REVIEWERS' COMMENTS

Reviewer #1 (Remarks to the Author):

This article has been comprehensively revised and improved according to the reviewer's comments and can be published in its current form.

Reviewer #2 (Remarks to the Author):

The authors faithfully addressed all of the minor concerns in the manuscript. Thus, I recommend this manuscript for publication as is.

Reviewer #3 (Remarks to the Author):

The revised manuscript and the comments by the authors answer my comments, and I consider the manuscript publishable to Nature Communications.

Response to the reviewers

Reviewer #1 (Remarks to the Author):

This article has been comprehensively revised and improved according to the reviewer's comments and can be published in its current form.

Reviewer #2 (Remarks to the Author):

The authors faithfully addressed all of the minor concerns in the manuscript. Thus, I recommend this manuscript for publication as is.

Reviewer #3 (Remarks to the Author):

The revised manuscript and the comments by the authors answer my comments, and I consider the manuscript publishable to Nature Communications.

Reply: Thank you very much for the strong supporting of our work.